# Impact of night and shift work on metabolic syndrome and its components: a cross-sectional study in an active middle-to-older-aged population-based sample

Virginie Bayon,[1] Mathieu Berger ![ORCID],[1] Geoffroy Solelhac ![ORCID],[1] José Haba-Rubio,[1] Pedro Marques-Vidal,[2] Marie-Pierre Strippoli,[3] Martin Preisig,[3] Damien Leger,[4,5] Raphael Heinzer[1]

VB and MB contributed equally.

VB and MB are joint first authors.

For numbered affiliations see end of article.

**Correspondence to**
Dr Mathieu Berger;
mathieuberger@outlook.com

## ABSTRACT

**Objectives** To examine the effects of work schedules on metabolic syndrome and its components in active middle-to-older-aged workers.

**Methods** A cross-sectional analysis including middle-to-older-aged active workers from the population-based CoLaus|PsyCoLaus study (Lausanne, Switzerland) was performed. Work schedule was self-reported and defined as follows: permanent day, day shift, night shift and permanent night work. Associations between work schedule and the risk of metabolic syndrome and its components were analysed using multivariable-adjusted logistic regressions.

**Results** A total of 2301 active workers (median age (IQR): 55.4 (50.8 to 60.4), 50.1% women) were included. Of these, 1905 were permanent day workers, 220 were day-shift workers, 134 were night-shift workers and 42 were permanent night-shift workers. There were significant interactions between sex and work schedule for metabolic syndrome, high triglycerides and visceral obesity. Men but not women permanent night workers had a higher prevalence of metabolic syndrome than permanent day workers in multivariable-adjusted analyses (OR 4.45 (95% CI 1.36 to 14.56)). Analysis of metabolic syndrome subcomponents showed that the association between work schedule and metabolic syndrome in men was mainly driven by visceral obesity (OR 3.35 (95% CI 1.04 to 10.76)). Conversely, women but not men working in night shift were at increased risk of having high triglycerides compared with permanent day workers (OR 2.92 (95% CI 1.03 to 8.27)).

**Conclusions** The risk of metabolic syndrome is higher in men working in permanent night shift compared with permanent day work, and this association could be mediated by visceral obesity.

## INTRODUCTION

Due to economic constraints, efficiency needs or performance objectives, night and shift work (3×8) has become highly prevalent in modern societies. Approximately, 18% of all

### Strengths and limitations of this study

► This study evaluated the effects of work schedules on metabolic syndrome and its subcomponent in a middle-to-older-aged general population setting with a precise and extensive assessment of cardiometabolic phenotypes.
► The association between different shift work schedules and metabolic syndrome was assessed after adjustment for multiple cofounders.
► Because the primary aim of the cohort was not to evaluate the impact of shift work, no precise characterisation of workstations and work rhythms (hourly amplitude, direction of rotation, duration of rotations and duration of exposition) was performed.
► A 'healthy worker effect' with a selection of 'night shift tolerant' workers cannot be ruled out given the older age of our sample.

European workers work in shifts, and this rate is as high as 35% in some countries.[1] Non-standard working schedules (eg, shift work, night work) are no longer limited to health and safety workers but are spread across all industries and services, from manufacturing, to transport, telecommunications and more.

Night and shift work interfere with the physiological circadian rhythm, desynchronising the biological clock, which can favour systemic inflammation.[2] Night and shift works are also associated with reduced and disturbed sleep.[3] Hence, both circadian disruption and short or poor sleep could be mediators explaining the relationship between night or shift work and chronic health conditions, including increased risk of cardiovascular and metabolic disorders.[4] Moreover, several laboratory-controlled studies showed that circadian rhythm desynchronisation and

**BMJ**

sleep restriction have detrimental effects on neuroendocrine, inflammatory and immune functions.[5]

The health-related impact of atypical work schedules has, thus, been a topic of interest for some time.[6] Sleep disturbances, decreased vigilance and increased risk of accidents are among the recognised short-term negative effects of night and shift work.[7] Longer term health effects have also been described and include increased risk of cardiovascular and metabolic disorders.[8 9] However, the impact of shift work on metabolic syndrome is not yet completely understood, particularly in the middle-to-older-aged population of workers though it is well established that the cardiometabolic risk gradually increases with advancing age.

Metabolic syndrome combines several interrelated metabolic risk factors associated with all-cause mortality.[10] Subjects with metabolic syndrome have a higher risk of cardiovascular disease mortality and morbidity.[11] Metabolic syndrome definition is based on five components: high blood pressure (BP), hyperglycaemic, high triglycerides, low–high-density lipoprotein (HDL) cholesterol and visceral obesity. A higher prevalence of metabolic syndrome and its components among night and shift workers has previously been suggested in some studies.[12 13] However, the specific effect of shift work and permanent night work remains largely unknown. Moreover, a recent systematic review concluded that there was insufficient evidence regarding the association between shift work and metabolic syndrome when confounding variables are taken into account.[14]

Thus, using data of active middle-to-older-aged workers from a population-based study, the aim of the present paper was to assess the cross-sectional association between metabolic syndrome and its components according to four types of work schedules (permanent day, day shift, night shift and permanent night-shift work).

## METHODS
### Study design
Cross-sectional analysis of a population-based cohort study.

### Population
CoLaus|PsyCoLaus is a population-based cohort exploring the biological, genetic and environmental determinants of cardiovascular risk factors, cardiovascular diseases and mental disorders in the middle-to-older-aged population of Lausanne, Switzerland. The methodological aspects (participant recruitment and follow-up) have been previously reported.[15] Briefly, a simple, non-stratified, random sample of 6734 subjects from the Lausanne population aged 35–75 years was recruited between 2003 and 2006. The baseline and three follow-up evaluations included physical and psychiatric examinations, blood sampling and self-completed questionnaires. All data analysed in the present paper were obtained from the second physical follow-up evaluation (n=4881), which took place between 2014 and 2017.

### Patient and public involvement
No patients or public were involved in this study design, conduct or analysis.

### Exposure and eligibility criteria
Professional activity and working hours were self-reported using the following questions: 'Are you currently engaged in a professional activity?'; 'What is your usual work schedule?' (day exclusively, rotation with no night work, rotation with night work, night work only). The number of work hours per week was also recorded. Participants not currently engaged in a professional activity were excluded from the present analysis. No other exclusion criteria were applied.

### Outcome assessment
Metabolic syndrome was defined according to the Joint Interim Statement[16] as the presence of at least three of the following five conditions: high BP (systolic BP ≥130 mm Hg or diastolic BP ≥85 mm Hg or use of antihypertensive medication); visceral obesity (waist circumference ≥88 cm in women or ≥102 cm in men); high triglycerides (≥1.7 mmol/L, or use of fibrates or nicotinic acid); low HDL–cholesterol levels (<1.30 mmol/L in women or <1.03 mmol/L in men or use of fibrates or nicotinic acid) and high fasting plasma glucose (≥5.6 mmol/L or use of antidiabetic medication). BP was measured three times on the left arm using an Omron HEM-907 (Matsusaka, Japan) automated oscillometric sphygmomanometer after at least a 10 min rest in the seated position. The mean of the last two measures was used. Venous blood samples were drawn after an overnight fast to measure the levels of glucose, HDL cholesterol, low HDL–cholesterol and triglycerides. Biological assays were performed at the clinical laboratory of the Lausanne university hospital within 2 hours of blood collection. Index of insulin resistance during fasting was assessed by the homeostatic model assessment of insulin resistance, calculated as the fasting insulin level (in milliunits per millilitre) times the fasting glucose level (in milligrams per litre) divided by 405. Waist circumference was measured two times with a non-stretchable tape over the unclothed abdomen at the mid-point between the lowest rib and the iliac crest. Hip circumference was also measured two times at the greater trochanters. For waist and hip, the mean of the two measurements was used and the waist-to-hip ratio was calculated.

### Covariates
The current socioprofessional category was self-reported by participants. Sociodemographic (age, sex) and lifestyle (smoking habit, alcohol intake, coffee consumption) data were collected by self-administered questionnaires. Educational level was categorised as *low* (primary), *middle* (apprenticeship or secondary school) or *high* (university). Smoking status was categorised as

*never, former* or *current.* Body weight and height were measured with participants standing without shoes in light indoor clothing. Body weight was measured in kilograms to the nearest 0.1 kg using a Seca scale (Seca, Hamburg, Germany). Height was measured to the nearest 5 mm using a Seca height gauge (Seca, Hamburg, Germany). Body mass index (BMI) was defined as weight (kg)/height$^2$ (m$^2$). Obesity was defined as BMI ≥30 kg/m$^2$.

Medication use was coded according to the WHO Anatomical Therapeutic Chemical Classification System (http://www.whocc.no/atcddd). Drugs influencing sleep included hypnotics or sedatives (N05C), anxiolytics (N05B) and antipsychotics (N05A). Diabetes was defined as fasting plasma glucose levels ≥7.0 mmol/L or use of antidiabetic medication.[17] Hypertension was defined as systolic BP ≥140 mm Hg and/or diastolic BP ≥90 mm Hg, and/or current use of antihypertensive medication.

The presence of a current major depressive disorder was retrospectively assigned according to Diagnostic and Statistical Manual of Mental Disorders, Fourth Edition criteria with information collected at the second and third psychiatric follow-up evaluation using the French translation of the semistructured Diagnostic Interview for Genetic Studies. Cardiovascular disease was defined as previous stroke, heart attack, coronary artery bypass grafting or percutaneous coronary intervention.

Subjective sleep characteristics were determined using the Pittsburgh Sleep Quality Index (PSQI),[18] the Epworth Sleepiness Scale (ESS)[19] and the Berlin questionnaire for sleep-disordered breathing (SDB).[20] Sleep quality was assessed with the PSQI and dichotomised into good/poor sleep quality (score ≤5/>5), and excessive daytime sleepiness was defined as an ESS score >10). A Berlin score ≥2 was defined as indicating a high risk of SDB.

Dietary intake was evaluated using a validated Food Frequency Questionnaire querying the consumption of 97 different food items, including portion size over the previous 4 weeks. The daily total energy intake was obtained as well as the proportion of macronutrients, alcohol and fibres.

Physical activity was evaluated with the physical activity frequency questionnaire (PAFQ).[21] The questionnaire lists 70 types of physical activity from various domains (eg, occupational, housework, leisure time, sports, etc) and participants indicated the number of days in the past week (0–7) and the duration per day (0–10 hour, in 15 min increments) for each activity. Energy expenditure corresponds to the sum of all the energy expenditure over 1 week divided by 7 to obtain a mean energy expenditure over a 24-hour period. Sedentary status was defined as spending more than 90% of daily energy in activities below moderate and high intensity (defined as requiring at least 4 times the basal metabolic rate. The percentage of total energy >4 metabolic equivalents was

also calculated to quantify moderate and high-intensity physical activity.

## Statistical analysis

Data distribution was graphically assessed using a normal Q-Q plot. Data were presented as number of participants (%) for categorical variables, mean±SD for normal distribution or median and IQR for non-normally distributed continuous variables. Univariate analyses of continuous data were performed using one-way Analyis of Variance (ANOVA) or Kruskal Wallis test follow by Bonferroni's post hoc or Tamhane's T2 as appropriate. Categorical variables were analysed using $\chi^2$ test or Fisher's exact test as appropriate. The associations between working schedules (permanent day, day-shift work, night-shift work and permanent night work) and metabolic syndrome (and its subcomponents) were determined using logistic regression analysis. Prior to this, the interaction of sex with the metabolic syndrome and each of its subcomponents was tested. In case of significant interaction, results were presented for both men and women, otherwise results were shown for the whole sample. Each cardiometabolic risk factor was first tested in univariate analysis (crude) then in two models with serial adjustment for potential confounders. Model one was adjusted for age (continuous), educational level (low, middle, high) and sex (except in case of significant sex×outcome interaction). Model 2: model 1 plus weekly alcohol consumption (continuous), smoking status (never, former, current) and BMI (normal weight, overweight, obese; except for visceral obesity). Model 3: model 2 plus daily total energy expenditure (continuous). Box-Tidwell tests were used to check the assumption of linearity for the logit of each covariate. If the assumption was violated, the square of the covariate was used or the covariate was transformed into categorical variable. To assess collinearity between covariates, a linear regression analysis including all covariates was performed, and the variance inflation factor (VIF) was calculated. A VIF ≤5 was considered as absence of multicollinearity. Results from logistic regression are presented as OR values with 95% CI. Permanent day workers were considered as the reference group.

All statistical analyses were performed using IBM SPSS Statistics V.26.0 for Macintosh (IBM Corp). Significant results were considered for a two-sided test with p<0.05.

## RESULTS
### Population characteristics

A total of 2301 participants were engaged in a professional activity at the second follow-up of the CoLaus|PsyCoLaus study. Among them, 1905 worked exclusively during the daytime (permanent day workers), 220 were rotation workers with no night work (day-shift workers), 134 were rotation workers with night work (night-shift workers) and 42 worked exclusively during the night (permanent night workers) (online supplemental figure 1).

Tables 1 and 2 show the baseline characteristics of the sample according to the four different work schedules. The mean age of the participants was 56.2±6.9 years and half of the sample (50.1%) were women. The proportions of men/women differed significantly according to work schedule: women were more likely to work in day shift and permanent night shift roles, while men were more likely to do night shift work. Mean BMI and waist circumference were significantly higher in night shift workers and permanent night workers compared with permanent day workers and day shift workers (p<0.001). Permanent night shift workers were more likely to smoke than other groups, whereas night-shift workers were less sedentary than their counterparts. Lipid levels and blood glucose analysis and sleep parameters in the different work schedule groups are also shown in tables 1 and 2.

### Prevalence of metabolic syndrome and its components according to work schedules

There were significant interactions between sex and work schedule for metabolic syndrome (p=0.009), high triglycerides (p=0.043) and visceral obesity (p=0.047), but not for high BP, high glucose and low HDL–cholesterol.

The prevalence of the metabolic syndrome was almost three times higher in men permanent night workers compared with men permanent-day workers; a similar trend was found for the prevalence of visceral obesity and low HDL-cholesterol (table 3). The prevalence of high glucose level in night shift workers and permanent night workers was nearly double that in permanent day workers (table 3).

### Association between metabolic syndrome and work schedules by patient sex

Compared with men permanent day workers, permanent night workers showed a higher risk of metabolic syndrome in univariate analysis (OR 6.48 (95% CI 2.40 to 17.46); online supplemental table 1). This significant association persisted after adjustment for age, educational level, alcohol consumption, smoking status and daily total energy expenditure (OR 4.45 (95% CI 1.36 to 14.56)) (figure 1). Conversely, the risk of metabolic syndrome in day shift-workers was lower than that in permanent day workers in crude analysis (OR 0.36 (95% CI 0.18 to 0.74)), and after adjustment in models 1 and 2 and 3 (online supplemental table 1). No significant association between work schedule and metabolic syndrome was found for women.

### Association of each component of metabolic syndrome with work schedule

In men, the risk of visceral obesity in permanent night workers was significantly higher than that in permanent day workers, including after adjustment for covariates (table 4). Moreover, the risk of elevated triglyceride levels in permanent night workers was increased in the crude analysis and after adjustment for age, educational level, alcohol consumption, smoking status and BMI (model 2)

but was no longer significant in the fully adjusted model 3 (table 4).

In women, night-shift workers showed a higher risk of elevated triglyceride levels, which persisted after multiple adjustments (table 4).

## DISCUSSION

In our middle-to-older-aged active general population sample, we found differential associations between permanent night work and the risk of metabolic syndrome for men and women. Indeed, permanent night work was only associated with a higher risk of metabolic syndrome in men but not in women. This association could be mediated by a higher risk of visceral obesity in men. The increased risk of metabolic syndrome is in line with previous studies.[22] Some studies even showed that the risk for the development of metabolic syndrome and each of its components gradually and independently increase with accumulated years of shift work.[23] Contrary to other studies, we found no association between permanent night work or night-shift work and metabolic syndrome in women.[24 25] In contrast to the findings on the metabolic syndrome as a whole, for the triglycerides component, we found an increased risk of elevated concentrations among shift workers in women but not in men. This supports previous evidence from Karlsson *et al* who also reported an elevated triglyceride level among shift workers in 60-year-old women.[26]

While the mechanisms underlying the observed increased risk of metabolic syndrome in shift or night workers have not been fully elucidated, several explanatory hypotheses can be proposed. First, sleep duration has been suggested to play a key role in the development of metabolic syndrome. A previous meta-analysis found that short sleep duration was significantly associated with a 27% increase in risk of metabolic syndrome, whereas long sleep duration was not.[27] Similar results were found in both men and women. In our study, self-reported sleep duration did not differ between the different groups of workers and, therefore, does not explain the increased risk of metabolic syndrome observed in permanent night workers among men. However, we cannot rule out that our findings might have been different if objective sleep duration measures were used because objective and subjective sleep duration can differ significantly. Unfortunately, objective sleep assessment could not be included in our analysis. Moreover, sleep fragmentation or an alteration of sleep structure due to irregular sleep schedule or circadian rhythm misalignment in night workers cannot be excluded and could be a possible explanation for the increased risk of metabolic syndrome.[28 29]

Second, dietary habits could contribute to development of the metabolic syndrome in night or shift workers, but available studies on this subject are scarce. A cross-sectional study comparing 98 rotating shift workers to 100 regular day workers demonstrated that total energy intake and contributions of macronutrients did not

**Table 1** Baseline characteristics according to work schedules

| | Permanent day workers (n=1905) | Day shift workers (n=220) | Night shift workers (n=134) | Permanent night workers (n=42) | p-value | N total |
|---|---|---|---|---|---|---|
| **Demographics and anthropometrics** | | | | | | |
| Age (years) | 55.0 (50.0–60.0) | 55.0 (50.5–59.5) | 54.5 (50.4–58.6) | 53.0 (48.8–57.2) | 0.070 | 2275 |
| Men, n (%) | 958 (50.3) | 89 (40.5) * | 88 (65.7) * | 18 (42.9) | **<0.001** | 2301 |
| Educational level, n (%) | | | | | **<0.001** | 2300 |
| Low | 791 (41.5) | 112 (50.9) | 71 (53.0) | 30 (71.4) * | | |
| High | 591 (31.0) * | 43 (19.5) | 23 (17.2) | 2 (4.8) | | |
| Body mass index (kg/m²) | 25.4 (22.6–28.5) | 25.5 (23.1–27.6) | 26.0 (23.2–30.0) † | 27.9 (25.4–31.3)†‡ | **<0.001** | 2228 |
| Waist circumference (cm) | 89.5 (81.0–98.5) | 89.0 (81.4–96.0) | 93.0 (84.3–102.0) †‡ | 95.0 (85.3–109.0)†‡ | **<0.001** | 2227 |
| Waist to hip ratio | 0.88±0.09 | 0.87±0.08 | 0.90±0.09 †‡ | 0.90±0.09 | **0.013** | 2227 |
| **Risk factors** | | | | | | |
| Metabolic syndrome, n (%) | 327 (17.2) | 25 (11.4) | 25 (18.7) | 17 (40.5)* | **<0.001** | 2301 |
| Number of metabolic risk factors‡ | 1 (0–2) | 1 (0–2) | 1 (0–2) | 2 (1–3)†‡ | **0.006** | 2301 |
| Current major depressive disorder, n (%) | 115 (7.9) | 19 (11.7) | 10 (9.6) | 4 (12.1) | 0.319 | 1756 |
| Hypertension, n (%) | 653 (34.9) | 80 (36.7) | 45 (33.6) | 12 (28.6) | 0.764 | 2263 |
| Diabetes, n (%) | 100 (5.4) | 8 (3.7) | 18 (13.6) * | 4 (9.5) | **<0.001** | 2231 |
| Dyslipidaemia, n (%) | 413 (22.5) | 51 (23.6) | 29 (22.1) | 16 (38.1) | 0.123 | 2226 |
| Sleep drugs, n (%) | 109 (5.7) | 14 (6.4) | 4 (3.0) | 4 (9.5) | 0.367 | 2301 |
| Cardiovascular disease, n (%)§ | 55 (2.9) | 12 (5.5) | 5 (3.7) | 3 (7.1) | 0.102 | 2291 |
| **Risk factors** | | | | | | |
| Smoking status, n (%) | | | | | **0.011** | 2246 |
| Former | 689 (37.0) | 86 (40.8) | 57 (43.8) | 9 (22.0) * | | |
| Current | 404 (21.7) | 44 (20.9) | 22 (16.9) | 18 (43.9) * | | |
| Alcohol (units/week) | 4 (1–9) | 3 (0–7) | 3 (0–7) | 2 (0–6) | **0.010** | 2162 |
| Coffee consumption, n (%) | | | | | 0.961 | 2222 |
| None | 186 (10.1) | 23 (11.0) | 13 (10.2) | 4 (9.8) | | |
| 1–3 cups/day | 1154 (62.6) | 134 (64.1) | 78 (60.9) | 28 (68.3) | | |
| ≥4 cups/day | 504 (27.3) | 52 (24.9) | 37 (28.9) | 9 (22.0) | | |
| Total energy intake (Kcals/day) | 1756±664 | 1761±654 | 1828±719 | 1853±619 | 0.603 | 1996 |

Continued

**Table 1** Continued

| | Permanent day workers (n=1905) | Day shift workers (n=220) | Night shift workers (n=134) | Permanent night workers (n=42) | p-value | N total |
|---|---|---|---|---|---|---|
| **Physical activity** | | | | | | |
| Total energy expenditure (Kcals/day) | 2656 (2297–3076) | 2698 (2336–3046) | 3118 (2735–3578)†‡ | 2663 (2356–3164) | <0.001 | 1828 |
| Activity ≥4 MET (% total activity) | 10.1 (1.9–18.4) | 8.8 (1.2–20.2) | 14.4 (4.9–25.3)† | 6.5 (0.3–16.1) | 0.005 | 1828 |
| Sedentary status, n (%) | 758 (49.2) | 92 (55.1) | 32 (34.4) * | 16 (57.1) | 0.011 | 1828 |
| **Blood analysis** | | | | | | |
| Total cholesterol (mmol/L) | 5.3±0.9 | 5.4±0.9 | 5.3±0.9 | 5.4±1.0 | 0.928 | 2226 |
| HDL cholesterol (mmol/L) | 1.5 (1.2–1.9) | 1.6 (1.3–1.9) | 1.4 (1.2–1.8) | 1.4 (1.1–1.7) | 0.013 | 2226 |
| LDL cholesterol (mmol/L) | 3.2±0.8 | 3.2±0.8 | 3.2±0.8 | 3.2±0.9 | 0.958 | 2226 |
| Triglycerides (mmol/L) | 1.0 (0.8–1.4) | 1.0 (0.8–1.5) | 1.1 (0.8–1.5) | 1.2 (0.9–1.8) | 0.278 | 2226 |
| Fasting glucose (mmol/L) | 5.2 (4.9–5.5) | 5.1 (4.8–5.5) | 5.2 (4.9–5.8) | 5.5 (5–5.9) | 0.026 | 2226 |
| Insulin (microIU/mL) | 7 (4.8–10.6) | 7.2 (4.6–10.9) | 7.3 (5.0–11.7) | 8.8 (6.5–12.9) | 0.027 | 2218 |
| HOMA-IR¶ | 1.6 (1.1–2.6) | 1.6 (1.0–2.6) | 1.7 (1.2–3.2) | 2.1 (1.5–3.4) | 0.012 | 2218 |

Data are presented as mean±SD or median and IQR for continuous variables and number of participants (%) for categorical variables. P-value<0.05 are shown in bold.

*Adjusted residual > | 2 |.

†Statistically different from 'day only'; ‡statistically different from 'shift work without night'.

‡Metabolic risk factor corresponded to the five risk factors which defined the metabolic syndrome according to the Joint Interim Statement[16]: systolic blood pressure ≥130 mmHg or diastolic blood pressure ≥85 mm Hg or use of antihypertensive medication; waist circumference ≥88 cm in women or ≥102 cm in men); triglycerides≥1.7 mmol/L, or use of fibrates or nicotinic acid; HDL-cholesterol <1.30 mmol/L in women or <1.03 mmol/L in men, or use of fibrates or nicotinic acid; and high fasting plasma glucose (≥5.6 mmol/L or use of anti-diabetic medication).

§Cardiovascular disease was defined by previous stroke, heart attack, coronary artery bypass grafting or percutaneous coronary intervention.

¶Index of insulin resistance during fasting was assessed by the homeostatic model assessment of insulin resistance (HOMA-IR), calculated as the fasting insulin level (in milliunits per millilitre) times the fasting glucose level (in milligrams per litre) divided by 405.

HDL, high-density lipoprotein; LDL, low-density lipoprotein; MET, metabolic equivalent of task.

**Table 2** Working and sleep characteristics according to work schedules

| | Permanent day workers (n=1905) | Day shift workers (n=220) | Night shift workers (n=134) | Permanent night workers (n=42) | p-value | N total |
|---|---|---|---|---|---|---|
| **Working characteristics** | | | | | | |
| Number of working hours/week | 38.0±14.7 | 38.7±15.2 | 43.1±18.1 | 38.0±15.2 | 0.260 | 2285 |
| Work time, n (%) | | | | | 0.397 | 2258 |
| Full-time | 1569 (83.8) | 181 (84.6) | 111 (86.0) | 39 (92.9) | | |
| <50% | 304 (16.2) | 33 (15.4) | 18 (14.0) | 3 (7.1) | | |
| Example of physical intensity at work, n (%) | | | | | **<0.001** | 2135 |
| Sedentary (sitting/driving) | 1409 (79.5) | 105 (51.2) | 66 (55.0) | 14 (37.8) | | |
| Pushing wheelbarrow | 283 (16.0) | 81 (39.5) | 40 (33.3) | 16 (43.2) | | |
| Unloading a truck without assist. | 81 (4.6) | 19 (9.3) | 14 (11.7) | 7 (18.9) | | |
| **Sleep and vigilance** | | | | | | |
| Epworth Sleepiness Scale score | 6 (4–8) | 5 (3–8) | 6 (4–9) | 5 (3–8) | 0.623 | 1786 |
| Excessive daytime sleepiness, n (%)* | 182 (12.1) | 20 (12.6) | 14 (14.0) | 3 (11.1) | 0.950 | 1786 |
| Poor sleep quality, n (%)† | 415 (31.5) | 46 (37.4) | 27 (32.1) | 7 (35.0) | 0.600 | 1542 |
| High risk of SDB, n (%)‡ | 321 (21.3) | 34 (21.0) | 29 (28.4) | 8 (27.6) | 0.323 | 1800 |
| Self-reported total sleep time (hour) | 6.9±1.0 | 6.8±0.9 | 6.9±1.0 | 7.1±1.3 | 0.507 | 1542 |

Data are presented as mean±SD or median and IQR for continuous variables and number of participants (%) for categorical variables.
P-value<0.05 are shown in bold.
*Excessive daytime sleepiness was defined by an Epworth Sleepiness Scale score >10.
†Poor sleep quality was defined by a Pittsburgh Sleep Quality Index score >5.
‡High risk of SDB was defined by a Berlin score >2.
SDB, sleep-disordered breathing.

differ between the two groups, except for saturated lipids (+10% in shift workers).[30] However, meal distribution was different in the two groups. Similar to other studies,[31 32] we failed to demonstrate a difference in food intake and macronutrients components between night shift workers or permanent night workers compared with permanent day workers. Available data from our study mean that, unfortunately, we cannot rule out the possibility that night shift workers may have had a different circadian distribution of food intake rather than an increase in total daily intake.[33]

Third, circadian rhythm desynchronisation could be a major contributor to the increased risk of metabolic syndrome among night and shift workers. Still, the underlying pathophysiological mechanisms of this association remain poorly understood. Some animal studies suggested that reduced melatonin production, due to circadian rhythm disruption, could be associated with a higher rate of metabolic syndrome.[34] Furthermore, Fonken et al hypothesised that exposure to light at night altered circadian organisation and affected metabolic parameters in mice.[35] Their results emphasised that even weak night lighting (5 lux) is sufficient to desynchronise food consumption and physical activity rhythms, which could explain the observed metabolic disorders.[34] In humans, Corbalan-Tutau et al reported a reduced daily amplitude in melatonin and cortisol circadian patterns associated with metabolic disturbances in women.[36] Unfortunately, we did not measure melatonin and cortisol to confirm these findings in our sample.

With regards to physical activity, we surprisingly found that night shift workers were more active than day shift workers and permanent day workers. This may be due to greater opportunities to perform a physical activity compared with other diurnal workers or to more physically active work among night shift workers, although this should be interpreted with caution due to limited agreement between estimates of activity obtained by PAFQ and those obtained from accelerometers.[37]

Finally, the higher risk of metabolic syndrome we observed in night shift workers may be explained by a vitamin D deficiency.[38] It has been shown that high levels

**Table 3** Prevalence of metabolic syndrome and its subcomponents according to work schedule

| | Permanent day workers (n=1905) | Day shift workers (n=220) | Night shift workers (n=134) | Permanent night workers (n=42) | p-value |
|---|---|---|---|---|---|
| **Metabolic syndrome** | | | | | |
| Men | 226 (23.6) | 9 (10.1) | 17 (19.3) | 12 (66.7) | **<0.001** |
| Women | 101 (10.7) | 16 (12.2) | 8 (17.4) | 5 (20.8) | 0.225 |
| High BP | 826 (43.4) | 91 (41.4) | 64 (47.8) | 23 (54.8) | 0.313 |
| High glucose | 472 (24.8) | 50 (22.7) | 47 (35.1) | 16 (38.1) | **0.010** |
| **High triglycerides** | | | | | |
| Men | 243 (26.2) | 25 (29.1) | 18 (21.2) | 11 (61.1) | **0.006** |
| Women | 86 (9.5) | 16 (12.3) | 9 (19.6) | 3 (12.5) | 0.183 |
| Low HDL-cholesterol | 201 (10.9) | 19 (8.8) | 10 (7.6) | 9 (21.4) | 0.064 |
| **Visceral obesity** | | | | | |
| Men | 220 (23.7) | 16 (18.6) | 23 (26.7) | 11 (61.1) | **0.002** |
| Women | 302 (33.3) | 55 (42.3) | 21 (45.7) | 11 (45.8) | 0.051 |

Data are presented as n (%).
Where there was an interaction of outcome*sex, results are presented separately for men and women, otherwise for the whole cohort.
BP, blood pressure; HDL, high-density lipoprotein.

of vitamin D among middle-aged and elderly populations are associated with a substantial decrease in cardiovascular disease, type 2 diabetes and metabolic syndrome.[39] Although we did not measure the vitamin D levels in our different groups of workers, we can hypothesise that permanent night workers have lower exposure to sunlight and may, therefore, be at higher risk of vitamin D deficiency.[40]

In our study, among the components of the metabolic syndrome, an elevated risk of visceral obesity was found in men permanent night workers. This finding is consistent with a recent meta-analysis, which found that shift workers had a higher frequency of abdominal obesity than other obesity types and permanent night workers

demonstrated a 29% higher risk of central obesity than rotating shift workers.[41]

The main strength of the present study is its large population-based sample of middle-to-older-aged workers with a precise and extensive assessment of cardiometabolic phenotypes. Indeed, previous studies were mainly performed in younger specific populations of workers or, in particular, sectors of activity, such as public health and emergency, which limit the generalisability to other types of shift or night work. In addition, most studies have assessed the risk of metabolic syndrome in shift workers compared with day workers, but few studies have differentiated between shift workers, permanent night workers and shift workers with and without night work.

There are also some limitations that need to be mentioned. First, this study had a cross-sectional design, which did not allow to assess causality but only cross-sectional associations that remain to be confirmed in prospective studies. Because the primary aim of the CoLaus|PsyCoLaus study was not to evaluate the impact of shift work, the questions related to shift work were only asked at the follow-up 2 (2014–2017), preventing us to investigate longitudinal associations. Moreover, no precise characterisation of workstations and work rhythms (hourly amplitude, direction of rotation, duration of rotations and duration of exposition) were performed. Likewise, it would have been interesting to have any information regarding food intakes or other habits in the workplaces. Second, a 'healthy worker effect' with a selection of 'night shift tolerant' workers cannot be ruled out given the older age of our sample. Third, our sample of permanent night workers is rather small but we may assume that workers move away from night shift work with advancing age due to poorer tolerability and less family

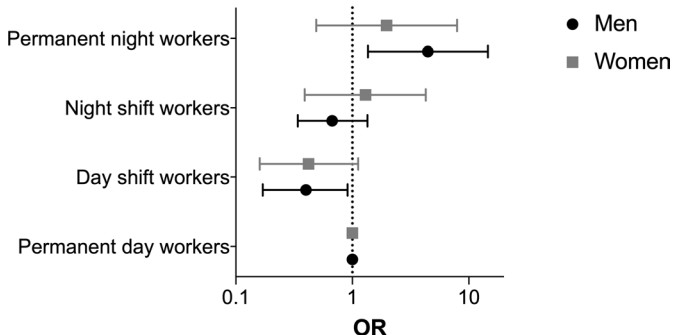

**Metabolic syndrome**

● Men
■ Women

**Figure 1** Multivariable-adjusted risk of metabolic syndrome according to work schedule and sex. Data are presented on a logarithmic scale and were analysed using multivariable logistic regression with adjustment for age, educational level, weekly alcohol consumption, smoking status and daily total energy expenditure (model 3).

**Table 4** Association of each component of the metabolic syndrome with work schedule

| | Crude | | Model 1 | | Model 2 | | Model 3 | |
|---|---|---|---|---|---|---|---|---|
| | OR (95% CI) | p-value | OR (95% CI) | p-value | OR (95% CI) | p-value | OR (95% CI) | p-value |
| **High BP** | n=2301 | | n=2274 | | n=2066 | | n=1731 | |
| Day shift-workers | 0.92 (0.69 to 1.22) | 0.572 | 1.05 (0.78 to 1.42) | 0.746 | 1.02 (0.73 to 1.42) | 0.907 | 1.06 (0.73 to 1.54) | 0.757 |
| Night shift-workers | 1.19 (0.84 to 1.70) | 0.321 | 1.02 (0.70 to 1.49) | 0.912 | 0.92 (0.61 to 1.39) | 0.682 | 1.01 (0.62 to 1.63) | 0.983 |
| Permanent night workers | 1.58 (0.86 to 2.92) | 0.144 | 1.78 (0.93 to 3.41) | 0.081 | 1.60 (0.77 to 3.31) | 0.204 | 1.90 (0.79 to 4.58) | 0.155 |
| **High fasting glucose** | n=2301 | | n=2274 | | n=2066 | | n=1731 | |
| Day shift-workers | 0.89 (0.64 to 1.25) | 0.504 | 1.05 (0.74 to 1.50) | 0.776 | 1.07 (0.73 to 1.58) | 0.735 | 1.04 (0.66 to 1.63) | 0.883 |
| Night shift-workers | 1.64 (1.13 to 2.37) | **0.009** | 1.36 (0.91 to 2.02) | 0.135 | 1.44 (0.93 to 2.24) | 0.106 | 1.26 (0.74 to 2.14) | 0.389 |
| Permanent night workers | 1.87 (0.99 to 3.51) | 0.052 | 2.14 (1.07 to 4.29) | 0.031 | 1.70 (0.79 to 3.64) | 0.173 | 1.31 (0.52 to 3.29) | 0.572 |
| **High triglycerides** | | | | | | | | |
| *Men* | n=1117 | | n=1116 | | n=1038 | | n=886 | |
| Day shift-workers | 1.16 (0.71 to 1.88) | 0.562 | 1.14 (0.70 to 1.87) | 0.593 | 1.30 (0.77 to 2.19) | 0.324 | 1.32 (0.73 to 2.40) | 0.360 |
| Night shift-workers | 0.76 (0.44 to 1.30) | 0.313 | 0.74 (0.43 to 1.28) | 0.287 | 0.86 (0.49 to 1.52) | 0.604 | 0.97 (0.52 to 1.84) | 0.936 |
| Permanent night workers | 4.43 (1.70 to 11.56) | **0.002** | 4.31 (1.64 to 11.30) | **0.003** | 3.50 (1.19 to 10.26) | **0.023** | 3.27 (0.99 to 10.77) | 0.051 |
| *Women* | n=1109 | | n=1105 | | n=1020 | | n=837 | |
| Day shift-workers | 1.34 (0.76 to 2.37) | 0.309 | 1.34 (0.75 to 2.38) | 0.320 | 1.19 (0.63 to 2.24) | 0.594 | 0.92 (0.41 to 2.03) | 0.828 |
| Night shift-workers | 2.33 (1.09 to 4.99) | **0.030** | 2.29 (1.06 to 4.95) | **0.035** | 2.65 (1.14 to 6.15) | **0.023** | 2.92 (1.03 to 8.27) | **0.044** |
| Permanent night workers | 1.37 (0.40 to 4.68) | 0.618 | 1.36 (0.39 to 4.73) | 0.625 | 1.09 (0.30 to 3.97) | 0.899 | 0.53 (0.06 to 4.32) | 0.549 |
| **Low HDL–cholesterol** | n=2226 | | n=2221 | | n=2058 | | n=1723 | |
| Day shift-workers | 0.79 (0.50 to 1.29) | 0.336 | 0.75 (0.46 to 1.23) | 0.255 | 0.62 (0.28 to 1.40) | 0.252 | 0.74 (0.39 to 1.39) | 0.348 |
| Night shift-workers | 0.67 (0.35 to 1.30) | 0.240 | 0.60 (0.31 to 1.18) | 0.138 | 0.58 (0.29 to 1.15) | 0.116 | 0.66 (0.30 to 1.45) | 0.300 |
| Permanent night workers | 2.22 (1.05 to 4.71) | **0.038** | 1.90 (0.89 to 4.08) | 0.099 | 1.61 (0.71 to 3.64) | 0.252 | 1.47 (0.52 to 4.18) | 0.468 |
| **Visceral obesity** | | | | | | | | |
| *Men* | n=1119 | | n=1118 | | n=1043 | | n=890 | |
| Day shift-workers | 0.74 (0.42 to 1.29) | 0.288 | 0.75 (0.42 to 1.34) | 0.333 | 0.84 (0.47 to 1.51) | 0.561 | 0.72 (0.36 to 1.42) | 0.341 |
| Night shift-workers | 1.18 (0.71 to 1.94) | 0.525 | 1.11 (0.66 to 1.84) | 0.704 | 1.06 (0.61 to 1.85) | 0.257 | 0.84 (0.44 to 1.63) | 0.612 |
| Permanent night workers | 5.06 (1.94 to 13.22) | **0.001** | 5.27 (1.99 to 13.98) | **0.001** | 4.79 (1.64 to 14.03) | **0.004** | 3.35 (1.04 to 10.76) | **0.042** |
| *Women* | n=1108 | | n=1104 | | n=1022 | | n=839 | |
| Day shift-workers | 1.47 (1.01 to 2.14) | **0.043** | 1.48 (1.01 to 2.17) | **0.043** | 1.31 (0.87 to 1.97) | 0.194 | 1.05 (0.65 to 1.70) | 0.852 |
| Night shift-workers | 1.70 (0.93 to 3.06) | 0.086 | 1.79 (0.98 to 3.29) | 0.059 | 1.91 (1.01 to 3.62) | **0.047** | 1.51 (0.66 to 3.10) | 0.324 |
| Permanent night workers | 1.70 (0.75 to 3.84) | 0.203 | 1.69 (0.73 to 3.92) | 0.219 | 1.75 (0.72 to 4.23) | 0.217 | 0.83 (0.23 to 2.99) | 0.971 |

Data are presented as OR and 95% confidence intervals (CI). For each component analysed, the 'permanent day workers' were considered as the reference group. p-values<0.05 are in bold. Model 1 was adjusted for age (continuous), age square (continuous), sex (except for sex subanalysis) and educational level (middle, low, high). Model 2 was additionally adjusted for weekly alcohol consumption (continuous), smoking status (never, former, current) and for BMI (normal weight, overweight, obese) (except for visceral obesity). Model 3 was additionally adjusted for daily total energy expenditure (continuous).

BMI, body mass index; BP, blood pressure; HDL, high-density lipoprotein.

constraints. Fourth, there were some missing data on self-reported sleep habits and diet parameters and, despite the use of validated questionnaires, declaration bias remains possible. Similarly, only self-reported physical activity was assessed in this study and it would have been interesting to have objective measures of physical activity and sleep to more accurately investigate their influence.

## CONCLUSION

Only men permanent night workers were at increased risk of metabolic syndrome compared with permanent day workers, and this association persisted after adjustment for sociodemographic confounders and daily total energy expenditure. From a clinical point of view, we advise monitor of not only BMI but also visceral obesity, particularly in men permanent night workers. Further prospective studies are needed to confirm theses cross-sectional results and elucidate the underline mechanisms.

**Author affiliations**
[1]Center of Investigation and Research on Sleep (CIRS), University Hospital of Lausanne (CHUV) and University of Lausanne (UNIL), Lausanne, Switzerland
[2]Department of Medicine, Internal Medicine, CHUV and University of Lausanne, Lausanne, Switzerland
[3]Department of Psychiatry, CHUV and University of Lausanne, Lausanne, Switzerland
[4]Centre du sommeil et de la vigilance, Hôtel Dieu, APHP, Paris, France
[5]EA 7330 VIFASOM, Sommeil-Vigilance-Fatigue et Santé Publique, Université Paris Descartes, Sorbonne Paris Cité, Paris, France

**Correction notice** This article has been corrected since it was first published. The lincese type has been updated to CC BY.

**Acknowledgements** The authors thank Prof. Mehdi Tafti, Prof. Vincent Mooser, Daniela Andries and Nadia Tobback for their important contribution to the HypnoLaus and CoLaus|PsyCoLaus Cohorts, the Lausanne population who volunteered to participate in the CoLaus|PsyCoLaus and HypnoLaus studies, as well as the whole team of CoLaus|PsyCoLaus.

**Contributors** VB, MB, PMV, MP and RH designed the study. JH-R, PM-V, M-PS, MP and RH collected the data. MB performed the statistical analysis. VB, MB, GS, JH-R, PMV, M-PS, MP, DL and RH interpreted the data. VB and MB wrote the first draft of the manuscript and GS, JH-R, PMV, M-PS, MP, DL and RH critically reviewed the manuscript. All authors undertake to give final approval of the version to be published and agree to be accountable for all aspects of the work. VB is the guarantor of this work and, as such, had full access to all the data in the study and takes responsibility for the integrity of the data and the accuracy of the data analysis.

**Funding** The HypnoLaus and the CoLaus/ PsyCoLaus studies were supported by research grants from GlaxoSmithKline ('not applicable'), the Faculty of Biology and Medicine of Lausanne ('not applicable'), the Swiss National Science Foundation (grants 3200B0-105993, 3200B0-118308, 33CSCO-122661, 33CS30-139468 and 33CS30-148401), Leenaards Foundation ('not applicable'), and Vaud Pulmonary League ('not applicable').

**Competing interests** None declared.

**Patient consent for publication** Consent obtained directly from patient(s)

**Ethics approval** This study involves human participants and was approved by the Institutional Ethics Committee of the University of Lausanne (decision reference 33/09) and written inform consent was obtained from all subjects. A copy of the written Inform Consent form was handed out to the subjects. A further copy was provided for the archives of the study in the Center for Investigation and Research in Sleep (CIRS, Lausanne University Hospital, Switzerland). Participants gave informed consent to participate in the study before taking part.

**Provenance and peer review** Not commissioned; externally peer reviewed.

**Data availability statement** Data may be obtained from a third party and are not publicly available. Due to the sensitivity of the data and the lack of consent for online posting, individual data cannot be made accessible. Only metadata will be made available in digital repositories. Metadata requests can also be made via the study website: www.colaus-psycolaus.ch.

**ORCID iDs**
Mathieu Berger http://orcid.org/0000-0002-6415-7694
Geoffroy Solelhac http://orcid.org/0000-0002-7135-133X

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
