## [Reviewer comments · BMJ Open]

ARTICLE DETAILS

TITLE (PROVISIONAL)	Impact of night and shift work on metabolic syndrome and its components: A cross-sectional study in an active middle-to-older-aged population-based sample
AUTHORS	Berger, Mathieu; Bayon, Virginie; Solelhac, Geoffroy; Haba-Rubio, José; Marques-Vidal, Pedro; Strippoli, Marie-Pierre; Preisig, Martin; Leger, Damien; Heinzer, Raphael

VERSION 1 – REVIEW

REVIEWER	Skogstad, Marit National Institute of Occupational Health, Occupational Medicine and Epidemiology
REVIEW RETURNED	30-Jul-2021

GENERAL COMMENTS	This is a well-written paper but due to the cross-sectional design, it could have some methodological problems. This is not sufficiently discussed in the present paper e.g. causality. The methods are not sufficiently addressed in the methods part. For instance, the reader is not supported information on the blood pressure apparatus type and model, nor analysis of the blood samples. There is no reason to include any information on depression/DSM-IV criteria in the paper. Please support the reader with definitions e.g. HOMA-IR, page 9. There are some linguistic challenges e.g. “our study demonstrates that only men permanent night workers..” should read “ only male...”, page 18 In Conclusion
---

REVIEWER	Moreno, Claudio University of Sao Paulo
REVIEW RETURNED	12-Oct-2021

GENERAL COMMENTS	Although the topic is interesting to be investigated, I felt the research question lacked a bit of novelty. There are many papers describing the odds of night workers developing metabolic syndrome. The present manuscript would add to the current knowledge if the data had information regarding the shifts. However, neither the start/end times nor the number of consecutive days of each shift is reported. Also, the authors say the data is a cohort but the analyses considered the data as a cross-sectional design. In my opinion, the analyses should be redone in order to check something new. Where is the analysis comparing the baseline data versus the data throughout the years? To know how much the odds increase per year would be quite new and innovative.
--

VERSION 1 – AUTHOR RESPONSE

REVIEWER # 1

Dr. Marit Skogstad, National Institute of Occupational Health

Comment 1 – Reviewer 1:

This is a well-written paper but due to the cross-sectional design, it could have some methodological problems. This is not sufficiently discussed in the present paper e.g. causality.

Response 1 – Reviewer 1:

We agree with the reviewer that cross-sectional design has some limitations and does not allow to assess causality. This was added in the limitations section page 17: “First, this study had a cross-sectional design which did not allow to assess causality but only cross-sectional associations that remain to be confirmed in prospective studies.”.

Comment 2 – Reviewer 1:

The methods are not sufficiently addressed in the methods part. For instance, the reader is not supported information on the blood pressure apparatus type and model, nor analysis of the blood samples.

Response 2 – Reviewer 1:

Methods section page 8 was revised and the requested information on blood pressure and blood samples we added: “Blood pressure was measured three times on the left arm using an Omron® HEM-907 (Matsusaka, Japan) automated oscillometric sphygmomanometer after at least a 10-min rest in the seated position.” / “Biological assays were performed at the clinical laboratory of the Lausanne university hospital within two hours of blood collection.”

Comment 3 – Reviewer 1:

There is no reason to include any information on depression/DSM-IV criteria in the paper.

Response 3 – Reviewer 1:

Although we understand the reviewer’s point, we believe that depression should be included in our analysis. Indeed, although, we did not find any difference between our groups, two systematic review and meta-analysis showed a bidirectional association between depression and metabolic syndrome, justifying to keep it in bivariate analysis (Pan et al. Bidirectional association between depression and metabolic syndrome. *Diabetes Care* 2012; Repousi et al. Depression and metabolic syndrome in the older population: a review of evidence. *Sleep Medicine* 2018).

Comment 4 – Reviewer 1:

Please support the reader with definitions e.g. HOMA-IR, page 9.

Response 4 – Reviewer 1:

Index of insulin resistance during fasting was assessed by the homeostatic model assessment of insulin resistance (HOMA-IR), calculated as the fasting insulin level (in milliunits per milliliter) times the fasting glucose level (in millimoles per liter) divided by 22.5. This was added in the method section page 9.

Comment 5 – Reviewer 1:

There are some linguistic challenges e.g. “our study demonstrates that only men permanent night workers.” should read “only male...”, page 18 In Conclusion

Response 5 – Reviewer 1:

We thank the reviewer for her suggestion. The paper was carefully proofread and the first sentence of the conclusion was corrected as suggested.

REVIEWER # 2

Dr. Claudio Moreno, University of Sao Paulo

Comment 1 – Reviewer 2:

Although the topic is interesting to be investigated, I felt the research question lacked a bit of novelty. There are many papers describing the odds of night workers developing metabolic syndrome. The present manuscript would add to the current knowledge if the data had information regarding the shifts. However, neither the start/end times nor the number of consecutive days of each shift is reported. Also, the authors say the data is a cohort but the analyses considered the data as a cross-sectional design. In my opinion, the analyses should be redone in order to check something new. Where is the analysis comparing the baseline data versus the data throughout the years? To know how much the odds increase per year would be quite new and innovative.

Response 1 – Reviewer 2:

We agree with the reviewer that a prospective design with information about shift schedules would have been ideal. Unfortunately, as mentioned in the manuscript, the Colaus-PsychoLaus-HypnoLaus cohort were not primarily designed to assess the impact of shift work. Therefore, information on shift work items were added only at the second follow-up of Colaus (2014-2017) which prevented us from performing a prospective analysis. Although the Colaus-PsychoLaus study is still ongoing and a third follow-up was performed between 2017 and 2021, data are not yet available and there was numerous loss to follow-up due to the Covid pandemic. Furthermore, our population is a rather old and we expect that many of them retired or stopped performing shift work at this third follow-up limiting the interest of this potential longitudinal analysis in the next years.

We apologize for the lack of clarity regarding the design of the study. This is now clarified in the title as well as in the methods and discussion.

We also believe that it is important to replicate results from previous studies in different populations from different area and origins to confirm their results. Moreover, considering that previous studies were performed up to 20 years ago, population and shift work patterns have likely evolved. Lastly, a recent systematic review concluded that there was insufficient evidence regarding the association between shift work and metabolic syndrome when confounding variables are taken into account (Canuto R, Garcez AS, Olinto MT. Metabolic syndrome and shift work: a systematic review. Sleep Med Rev 2013;17:425-31.), making it worthwhile, in our opinion, to replicate the results.

VERSION 2 – REVIEW

REVIEWER	Moreno, Claudio University of Sao Paulo
REVIEW RETURNED	30-Nov-2021

GENERAL COMMENTS	This is a cross-sectional study on metabolic syndrome and its association with night shift work. This is not new in the literature but the focus on middle-age population is original. Thus, it is quite surprising that there is nothing about age in the introduction. Also, there is no comments about the age of participants in the methods session in the abstract. Thus, the introduction should include a
---

	rationale about age as a relevant factor to be studied in the context of shift work and metabolism. The abstract has to include the age range of recruitment. The highlights of limitations and strengths of this study reflect a little bit of this confusion. This study does not evaluated "the effects of work schedulesin a general population...". The study was conducted only with a very specific age range. This is, in fact, what makes the study interesting. The relevance of ageing and its relation with metabolic syndrome must be shown in the discussion as well. I have a concern regarding the sample size of the permanent night work group. Why was this group so small? It is clear that the main finding of the study comes from this group, so we should know more about it. In the discussion, the authors say that a possible explanation for finding high risk for MS among permanent night workers could be lack of vitamin D. However, some studies showed that these workers are more exposed to sunlight during the day compared to shift workers. Please, comment on that. I also would like to see a comment in the discussion regarding the possible workplaces of the studied people and its relation with food intake and other habits (does any workplace gives meals to the employees?). In addition, the discussion should include more about food intake and physical activity.
--	---

VERSION 2 – AUTHOR RESPONSE

REVIEWER # 2

Dr. Claudio Moreno, University of Sao Paulo

Comment 1 – Reviewer 2:

This is a cross-sectional study on metabolic syndrome and its association with night shift work. This is not new in the literature but the focus on middle-age population is original. Thus, it is quite surprising that there is nothing about age in the introduction. Also, there is no comments about the age of participants in the methods session in the abstract. Thus, the introduction should include a rationale about age as a relevant factor to be studied in the context of shift work and metabolism. The abstract has to include the age range of recruitment. The highlights of limitations and strengths of this study reflect a little bit of this confusion. This study does not evaluated "the effects of work schedulesin a general population...". The study was conducted only with a very specific age range. This is, in fact, what makes the study interesting. The relevance of ageing and its relation with metabolic syndrome must be shown in the discussion as well.

Response 1 – Reviewer 2:

We thank the reviewer for his relevant comment. It is true that the middle-to-older-aged characteristics of our population was not enough emphasized in the previous version of the manuscript. The suggested changes were incorporated in this R2 version of the manuscript: age characteristics and age range were added in the abstract, in the introduction and in the discussion.

Comment 2 – Reviewer 2:

I have a concern regarding the sample size of the permanent night work group. Why was this group so small? It is clear that the main finding of the study comes from this group, so we should know more about it.

Response 2 – Reviewer 2:

Several hypotheses may explain why the sample size of permanent night workers was rather small. First, we may assume that workers tend to move away from night shift work with advancing age due to poorer tolerability. Second, after 50 years of age some workers are allowed to stop shifts in some work areas in Switzerland. Third, permanent night work is not the main shift work pattern in active workers but concerns mainly specific population of workers such as emergency workers.

Comment 3 – Reviewer 2:

In the discussion, the authors say that a possible explanation for finding high risk for MS among permanent night workers could be lack of vitamin D. However, some studies showed that these workers are more exposed to sunlight during the day compared to shift workers. Please, comment on that.

Response 3 – Reviewer 2:

Although we may hypothesize, as suggested by the reviewer, that permanent night workers sleep less and have more time to perform outdoor activities during the day, we did not find any studies in the literature supporting this hypothesis. In contrast, we found one study supporting our hypothesis of lack of vitamin D (Daugaard et al. Indoor, outdoor, and night work and blood concentrations of vitamin D and parathyroid hormone. Scand J Work Environ Health 2018). This study examined the blood levels of 25-hydroxyvitamin D (25OHD) in 425 workers and showed that permanent night workers had lower levels of 25OHD and a higher probability of vitamin D insufficiency. The reference was added in the manuscript page 17.

Comment 4 – Reviewer 2:

I also would like to see a comment in the discussion regarding the possible workplaces of the studied people and its relation with food intake and other habits (does any workplace gives meals to the employees?). In addition, the discussion should include more about food intake and physical activity.

Response 4 – Reviewer 2:

This is a very good point and indeed, being able to have a quality meal during shift or night work probably has a great influence on weight control and the risk of developing metabolic syndrome. Unfortunately, we do not have information about participants' locations and therefore could report information regarding their food intake or other habits. We have added this point within the limits of the study.

VERSION 3 – REVIEW

REVIEWER	Moreno, Claudio University of Sao Paulo
REVIEW RETURNED	23-Dec-2021
GENERAL COMMENTS	The authors have answered my comments properly.